# Arrhythmic Sudden Cardiac Death and the Role of Implantable Cardioverter-Defibrillator in Patients with Cardiac Amyloidosis—A Narrative Literature Review

**DOI:** 10.3390/jcm10091858

**Published:** 2021-04-25

**Authors:** Aleksandra Liżewska-Springer, Grzegorz Sławiński, Ewa Lewicka

**Affiliations:** Department of Cardiology and Electrotherapy, Medical University of Gdańsk, 80-210 Gdańsk, Poland; kardio2@gumed.edu.pl (G.S.); elew@gumed.edu.pl (E.L.)

**Keywords:** cardiac amyloidosis, implantable cardioverter-defibrillator, sudden cardiac death, ventricular arrhythmias

## Abstract

Cardiac amyloidosis (CA) is considered to be associated with an increased risk of sudden cardiac death (SCD) due to ventricular tachyarrhythmias and electromechanical dissociation. However, current arrhythmic risk stratification and the role of an implantable cardioverter-defibrillator (ICD) for primary prevention of SCD remains unclear. This article provides a narrative review of the literature on electrophysiological abnormalities in the context of ventricular arrhythmias in patients with CA and the role of ICD in terms of survival benefit in this group of patients.

## 1. Introduction

Amyloidosis is a rare systemic disease characterized by the extracellular deposition of pathological insoluble fibrillar protein, known as amyloid, within various organs (mainly the heart and kidneys). The most common types of cardiac amyloidosis (CA) are caused by immunoglobulin-derived light chains (AL) and the precursor protein transthyretin (ATTR). Cardiac involvement occurs up to 60% patients, more commonly in AL amyloidosis and results in worse prognosis [1]. Therefore, increased clinician awareness and early CA diagnosis is crucial to improve outcomes. CA should be suspected particularly in patients with heart failure with preserved left ventricular ejection fraction (HFpEF) presenting “red flag” signs such as (1) either symmetrical or asymmetrical unexplained left and right ventricular hypertrophy with concomitant diastolic dysfunction and reduced global longitudinal left ventricular strain (LV GLS) with an “apical sparing” pattern in echocardiography, (2) discrepancy between the LV wall thickness and QRS voltage and the presence of pseudo-infarct pattern in electrocardiography, especially if associated with increased levels of N-terminal pro B-type natriuretic peptide (NT-proBNP [2]. Cardiac magnetic resonance imaging, endomyocardial biopsy and nuclear imaging play an important role in CA diagnosis. [2]. The treatment depends on the type of amyloidosis.

Chemotherapy and hematopoietic stem cell transplantation is primarily aimed at managing a clonal plasma cell dyscrasia in AL amyloidosis. On the contrary, chemotherapy plays no role in the treatment of ATTR CA. Clinical studies on various therapeutic agents that modify/inhibit amyloid fibril formation or stabilize mutant transthyretin (TTR) fibers are in progress [3,4]. Tafamidis, a TTR tetramer stabilizer, is the most extensively studied medication that showed the reduction in all-cause mortality and hospitalization rates in ATTR CA, especially if applied in early stages of the disease [5]. Apart from tetramer stabilizers, gene silencing drugs that interfere with the production of an abnormal form of TTR have been investigated. Among them, patisiran have led to the reversal of structural changes in the myocardium. Doxycycline, a tetracycline antibiotic, can potentially interfere with amyloid fibril formation through an unknown mechanism enhancing LV mechanical function [5].

Although novel treatment vastly improved survival in AL and ATTR cardiac amyloidosis, cardiovascular events account for more than two-thirds of fatal casualties in both groups [6]. Moreover, sudden cardiac death (SCD) accounts for up to 50% of all cardiac deaths [7]. Electromechanical dissociation is thought to be the most common cause of SCD in patients with cardiac amyloidosis; however, ventricular arrhythmias and conduction abnormalities are also common [8]. To date, a number of factors have been described, indicating an increased risk of overall mortality. However, little is known about risk factors for ventricular tachyarrhythmia’s as a cause of SCD in patients with amyloidosis and cardiac involvement. Therefore, identification of patients with CA who may be eligible for implantable cardioverter-defibrillator (ICD) is challenging. It remains unclear whether ICD prevents SCD in these patients.

## 2. Materials and Results

We performed a narrative review, rather than a systematic review of the literature, that focuses on arrhythmic sudden cardiac death and the role of ICD in patients with CA. We used PubMed and Google Scholar to find articles published in the years 1997–2020. We searched using the following keywords or search phrases: “cardiac amyloidosis and implantable cardioverter-defibrillator,” “amyloidosis and SCD,” “prevention of SCD in cardiac amyloidosis,” “electrophysiology in cardiac amyloidosis,” “ventricular arrhythmia and cardiac amyloidosis”. A total of 1734 potentially-relevant records were identified. After screening the titles and abstracts, 11 records were selected for a detailed analysis (10 full-texts and 1 abstract). Finally, the references of all analyzed articles were screened for relevant papers not found in the initial search.

## 3. Prognostic Factors and Electrophysiological Abnormalities in Patients with Cardiac Amyloidosis

Cardiac involvement is the determinant of prognosis in CA. Risk of death in patients with AL amyloidosis can be stratified using the revised Mayo staging models, including cardiac biomarkers: serum troponins (cTnT ≥ 0.025 ng/mL), NT-proBNP (≥1.800 pg/mL) and serum immunoglobulin free light chain difference (FLC-diff) ≥ 18 mg/dL) [9]. Kumar et al. assigned one point for each of these abnormalities [9]. Their median overall survival from diagnosis was 94.1, 40.3, 14, and 5.8 months, respectively. A European collaborative study additionally reported that very high NT-proBNP levels (>8500 pg/mL) indicate patients at very high risk with a median overall survival of only 3 months [10]. Lilleness at al. demonstrated that easier and more accessible prognostic scoring system, the Boston University staging system, including BNP (>81 pg/mL) and cTnI (>0.1 ng/mL), also accurately identified cardiac involvement and stratified overall survival [11]. Similar to AL amyloidosis, a staging system including markers of increased myocardial stress, such as NT-proBNP and high sensitivity cTnT, has been proposed for ATTR amyloidosis. Patients with both: cTnT > 0.05 ng/mL and NT-proBNP > 3.000 pg/mL had the worst prognosis with a median survival of only 20 months [12]. Additional to elevation in cardiac biomarkers, renal dysfunction was also identified as a significant risk factor of worse prognosis. In the recently proposed prognostic system for staging ATTR amyloidosis, patients with decreased estimated glomerular filtation rate (eGFR < 45 mL/min/1.73 m^2^) and NT-proBNP > 3.000 pg/mL had significantly worse survival compared to those not meeting these cut-off values [13].

However, the mentioned staging models only predict overall mortality. Moreover, there is no significant correlation between cardiac biomarkers levels and the risk of ventricular arrhythmias [14].

Risk assessment of arrhythmic SCD in cardiac amyloidosis is still not well-defined. Understanding the pathophysiology of ventricular arrhythmias (VA) in CA is crucial to predict the risk of death. Amyloid in the extracellular spaces distorts the myocardial cells and can also infiltrate cardiac conduction system and coronary arteries. Besides infiltration, amyloidogenic light chains in AL amyloidosis may directly impair cardiomyocyte function through an increase in cellular oxidant stress. It appears that myocardial scarring and fibrosis that are typical of chronic ischemic or non-ischemic cardiomyopathies are less common in CA. Among imaging studies, cardiac magnetic resonance (CMR) plays an important role not only in the diagnosis of CA but also provides important prognostic information. In amyloidosis, CMR enables myocardial tissue characterization by means of T_1_- and T_2_-weighted imaging sequences, T_1_ mapping (pre- and post-contrast), late gadolinium enhancement (LGE) and extracellular volume (ECV) imaging. Global subendocardial or transmural pattern of LGE, and to a lesser degree, a focal patchy LGE, are all features of CA. LGE has been recognized as a marker of amyloidogenesis and fibrosis. The extent of LGE may also serve as a surrogate of arrhythmogenic substrate for the occurrence of ventricular arrhythmias [2,15]. The two-year survival in CA patients without LGE was 92%, whereas it was significantly lower in those who showed subendocardial or transmural LGE (81% and 45%, respectively) [16]. Both in AL and ATTR cardiac amyloidosis, the presence of transmural LGE has been shown to be an independent predictor of worse survival [16]. However, the limitation of LGE is that it is difficult to quantify, making it difficult to track changes in CA, e.g., due to treatment. This has been overcome with the use of the technique of T_1_ mapping, which showed that native T_1_ values (pre-gadolinium contrast) are markedly higher in regions of amyloid deposition (or diffuse fibrosis). Post-contrast T_1_ mapping following gadolinium administration enables estimation of ECV. The ECV values are significantly elevated in CA and ECV is a robust marker of prognosis in CA. Moreover, the assessment of ECV as well as native T_1_ values enables tracking the disease over time and response to therapy. Additionally, T_2_ mapping provides data on T_2_ relaxation times which represent a myocardial edema and active inflammation and is potentially linked with arrhythmogenic potential. However, data on T_2_ mapping in CA are scarce so far. In a recent study [17], the presence of myocardial edema was shown in CA, as indicated by increased T_2_ relaxation times in patients with amyloidosis compared to control subjects and in untreated AL amyloidosis compared with treated AL and ATTR amyloidosis. In this study, T_2_ was a predictor of prognosis in AL amyloidosis, which may suggest mechanisms additional to amyloid infiltration contributing to mortality in this disease. The cause and mechanisms of ventricular arrhythmias in CA, however, are poorly understood and are likely to be multifactorial [18,19].

To better understand the underlying pathophysiology, Orini et al. combined the assessment of the electrophysiological and structural ventricular substrate from 21 CA patients (11 AL and 10 ATTR) [20]. The authors used a special electrocardiographic system with 256 electrodes for non-invasive epicardial mapping of ventricular potentials and cardiac magnetic resonance (CMR) imaging. When compared with healthy volunteers, patients with CA had significantly lower epicardial signal amplitude, slower and heterogeneous intraventricular conduction and prolonged and more spatially dispersed repolarization. Moreover, epicardial signal fractionation and average repolarization time increased with extracellular volume calculated in CMR. A strong inverse correlation was found between epicardial signal amplitude and native T1 in CMR. Both epicardial conduction and repolarization abnormalities were more notable in patients with AL amyloidosis compared with ATTR. Spatial conduction-repolarization heterogeneity is thought to be a marker of increased propensity to VA and sudden arrhythmic death in patients with heart failure and may contribute to higher mortality in AL amyloidosis [21]. This study also suggests a link between conduction-repolarization delay and increased extracellular deposition.

Invasive electrophysiological study (EPS) is infrequently performed in CA patients, and we found only two studies determining the spectrum of electrophysiological abnormalities among CA patients in EPS. Reisinger at al. demonstrated a prolongation of the His-ventricular (HV) interval >55 ms in the majority of the examined population (23 of 25 patients with AL amyloidosis confirmed in biopsy), which indicated disease of the distal His-Purkinje system [7]. Markedly prolonged HV interval (≥80 ms) was the only independent predictor for SCD in the multivariate analysis. The authors concluded that prolongation of the HV interval does not only indicate a risk of complete atrio-ventricular block due to the conduction system infiltration with amyloid fibrils and bradyarrhythmia as a potential cause of death, it may also indicate severe myocardial infiltration and serve as a marker of the propensity for lethal VA or acute electromechanical dissociation. Interestingly, in this study, monomorphic ventricular tachycardia (VT) was induced only in four patients during programmed ventricular stimulation, and similarly to other non-ischemic cardiomyopathies, VT non-inducibility showed little prognostic value.

In a study of 18 CA patients, Barbhaiya at al. demonstrated a prolonged HV interval >55 ms in all patients, which was more significant in those with ATTR amyloidosis (14 patients) [22]. Additionally, CA patients with concomitant atrial fibrillation (AF) or atrial tachycardia had larger areas of low voltage, as revealed by detailed left atrium mapping compared to age-matched controls of patients with persistent AF. Of the six patients who underwent programmed ventricular stimulation, two patients had induced monomorphic VT and received an ICD. However, the authors did not evaluate the effect of their findings on mortality.

## 4. The Role of ICD Therapy

Whether there is a selected population of patients with CA at risk of arrhythmic SCD (versus SCD due to electromechanical dissociation) who would benefit from ICD placement is still a matter of debate [23]. No robust predictors for malignant ventricular arrhythmias have been identified so far [24]. Due to insufficient data, the European Society of Cardiology consensus statement from 2015 does not provide recommendations on preventive ICD implantation in CA patients [25]. According to the Heart Rhythm Society (HRS) consensus statement from 2019, a prophylactic ICD may be considered in patients with non-sustained ventricular tachycardia (NSVT) in the course of AL and expected survival longer than one year [26]. However, it is only the class IIb recommendation.

We found 11 retrospective analyses of outcomes in patients with CA implanted with ICD for primary and secondary prevention. Table 1 summarizes the data available about the 720 patients reported to date. Almost a quarter of patients received appropriate ICD therapy and 88% of them survived immediately after device intervention. The incidence of inappropriate ICD interventions was low (7%). However, only 22% of patients who received appropriate ICD therapy survived the follow-up (data based on six publications), and in 68% of patients from the entire analyzed CA population, the ICD had probably no effect on their survival (Table 1). Kristen et al. indicated low efficacy of ICD therapy and emphasized the frequent occurrence of electromechanical dissociation in CA patients [27]. On the contrary, several of the analyzed studies reported frequent and successful ICD therapy, but none of these have demonstrated a survival benefit [14,28,29,30,31,32,33].These discrepancies are due to multiple limitations of these studies. First, all of the studies included in this review were retrospective with various sample sizes (3–472 patients). Second, the reports included patients with a variety of CA etiologies, including AL (0–100% of patients in these studies) and ATTR amyloidosis, while each has a different clinical presentation, natural history, prognosis and treatment. Regarding the prognosis, Harmon et al. indicated in the multivariate analysis that AL amyloidosis was an independent predictor of high mortality in CA patients [28]. Third, the majority of ICDs were implanted in primary prevention to patients who had left ventricular ejection fraction (LVEF) >35%. Qualification was mostly based on arbitrarily adopted criteria, which included the presence of different types of ventricular arrhythmias (such as NSVT or frequent premature ventricular beats) and/or non-postural syncope. Additionally, in four reports, the criteria for primary prevention were not specified, including the largest study involving 472 patients [34]. The predictive value of the above-mentioned criteria for primary SCD prevention is controversial.

Non-sustained VT is a common finding among patients with CA, and its role in predicting SCD in this population is debated, as it appears to have little discriminative value to identify those who die from VA [28,30]. In the meta-analysis by Halawa et al., despite high prevalence of NSVT (in 51% of CA patients), only 18% received appropriate ICD therapy [35]. Nevertheless, NSVT in the early stage of AL amyloidosis can be considered an indication for ICD implantation in primary prevention (class IIb) [26].

Unexplained syncope is a common and non-specific symptom in CA population and it can result from other causes than conduction disturbances or VA, such as orthostatic hypotension, autonomic dysfunction or the use of diuretics or vasodilating drugs [36]. In some patients, the qualification for primary ICD implantation is based on the standard left ventricular (LV) systolic dysfunction with LVEF ≤ 35%. However, in the majority of CA patients, the decline of LV systolic function is a late manifestation; therefore, other echocardiographic parameters are needed to assess LV function (even in those with preserved LVEF). Harmon et al. showed that VA were more common in patients with reduced LV two-dimensional global longitudinal strain (2D-GLS ≥ −15%) assessed by speckle-tracking echocardiography (STE) [28].

In studies included in this review, appropriate ICD therapies were found at different rates (in 6–100% of CA patients). Moreover, ICD programming in primary prevention (both detection and therapy) was not reported in these studies; however, this may be important in determining the actual ventricular arrhythmic burden in patients with CA. In addition, the programmed basic pacing rate was not clearly defined, and a higher percentage of right ventricular pacing (>40%) has been found to increase mortality among patients with CA [37]. Finally, most reports did not specify the time after which ICD was implanted after CA diagnosis. Meanwhile, it is very important as patients with CA are often diagnosed late in various stages of amyloidosis, and overall mortality is higher in advanced stages of the disease. Noteworthy, patients with greater cardiac involvement, manifested as higher NT-proBNP concentration and lower LVEF, may be at higher risk of death from electromechanical dissociation [14]. Therefore, ICD implantation can be more beneficial in those with cardiac involvement but with lower NT-proBNP level and preserved LVEF [30].

It is known that the risk of ventricular tachyarrhythmias and SCD varies with the type of CA and is significantly greater in patients with AL amyloidosis [23]. Based on the current state of knowledge, there are no robust guidelines for the decision to implant an ICD for primary prevention in CA. We propose an algorithm for ICD implantation in patients with AL amyloidosis and LVEF >35% (Figure 1). We believe that ICD should be considered in patients with registered NSVT and in early stages of the disease with less impairment in cardiac function, as indicated by minimally to moderately raised cardiac biomarkers. As NSVT has been documented to be a poor predictor of SCD in AL amyloidosis, the incidence of syncope, the decrease of LV-GLS in echocardiography and the presence of transmural LGE in CMR can further improve stratification.

In summary, because of several limitations of previous studies, the role of ICD in CA patients is controversial. Future randomized studies with larger sample sizes, strictly defined indications for primary prevention and defined ICD programming (VT detection and therapy) are required to draw final conclusions on ICD therapy for patients with CA.

## 5. Future Perspective of Studies on SCD Risk in Patients with Cardiac Amyloidosis

In the available literature, apart from the three studies mentioned [7,20,22], data on the spectrum of electrophysiological abnormalities and arrhythmogenic substrate in patients with CA are lacking. More data on risk factors of arrhythmic SCD are needed. Patients with CA can manifest symptoms of autonomic dysfunction, which is a hallmark feature of hereditary ATTR amyloidosis [38]. Whether cardiac autonomic dysfunction may trigger and maintain VA in CA in unknown. Reyners at al. showed that low heart rate variability was a strong predictor of one-year mortality in patients with AL amyloidosis [39]. Studies demonstrated a positive correlation of baroreflex sensitivity (BRS) on innervation density in patients with AL amyloidosis. Low values of innervation density, in turn, were associated with significantly poorer survival in this group of patients [40]. However, the usefulness of the assessment of various parameters reflecting autonomic system activity in predicting SCD has been questioned [16].

The T-wave alternans (TWA) phenomenon that was once considered to be a predictor of total mortality and SCD risk in patients with heart failure has not yet been studied in patients with CA [41]. Preliminary (unpublished) data on the TWA testing in six patients with AL amyloidosis in our group showed a normal (negative) result in all of them, as well as normal values of BRS; this research is currently ongoing.

## 6. Conclusions

Cardiac amyloidosis carries a high risk of SCD. However, ICD implantation for primary prevention of SCD remains controversial, and it is not clear whether ICDs improve survival in CA. Little data are available on the arrhythmic SCD risk stratification and the usual approach in these patients is secondary prevention or extrapolation of risk factors from other cardiomyopathies, e.g., impaired LV systolic dysfunction or NSVT with or without syncope. CA patients are treated with appropriate ICD settings similarly to other groups of patients. However, their overall mortality is still high, particularly in those in advanced stages of the disease, and thus, they are at increased risk of death due to electromechanical dissociation. Little is known about a potential arrhythmogenic substrate in CA patients, which may be different in AL and ATTR amyloidosis. Therefore, the challenge is to identify an at-risk patient in the early stage of the disease when VA risk predominates and who may benefit from ICD therapy. Further prospective studies are needed to understand the pathophysiology of cardiac arrhythmias in CA patients, including EPS with endocardial mapping and modern CMR imaging, to indicate predictors of arrhythmic SCD and finally define the role of ICD in this group of patients.

## Figures and Tables

**Figure 1 jcm-10-01858-f001:**
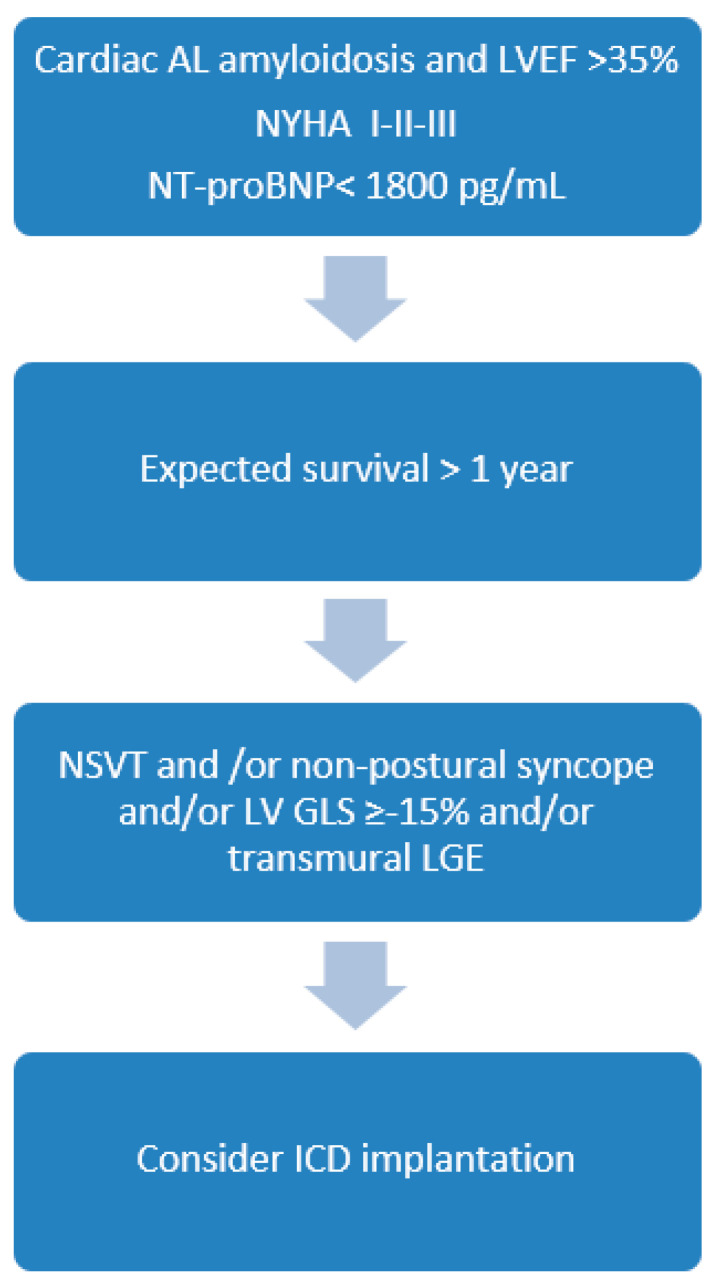
Proposed algorithm for qualifying patients with cardiac AL amyloidosis for ICD implantation in primary prevention. LVEF—left ventricular ejection fraction, T-proBNP—N-terminal pro B-type natriuretic peptide; NYHA—New York Heart Association; NSVT—non sustained ventricular tachycardia, LV-GLS—left ventricular global longitudinal strain assessed by speckle tracking echocardiography, LGE—late gadolinium enhancement in cardiac magnetic resonance imaging.

**Table 1 jcm-10-01858-t001:** Studies reporting on treatment with implantable cardioverter-defibrillator and survival in patients with cardiac amyloidosis.

Study/Year Published	CA Patients with ICD (n)	AL Amyloi-dosis	ICD in Primary Prevention (n/%)	Criteria for ICD Implantation in Primary Prevention	LVEF	History of Syncope (n/%)	Appropriate ICD Therapy (n/%)	Inappropriate ICD Therapy (n/%)	Survival Directly Post-ICD Therapy (n/%)	Survival during Follow-Up after Appropriate ICD Therapy (n/%)	Overall Survival (n/%)	Follow-Up Duration
Kristen et al. (2008) [27]	19	19/19 (100%)	19 (100%)	Syncope and/or frequent PVBs	≤45% in 5 pts	4 (21%)	2 (11%)	2 (11%)	1/2 (50%)	1/2(50%)	10/19 (53%)	811 ± 151 days
Kojima et al. (2012) [31]	3	3/3 (100%)	2 (67%)	NSVT	>55%	267%)	3 (100%)	0	3/3 (100%)	0/3 (0%)	0/3 (0%)	7 months (median)
Lin et al. (2013) [30]	53	33/53 (62%)	41 (77%)	LVEF ≤35% or syncope or NSVT	48 ± 17%	UN	15 (28%)	6 (11%)	UN	UN	21/53(40%)	23.25 ± 21.45 months
Varr et al. (2014) [14]	19	15/19 (79%)	15 (79%)	Not specified	≤45% in 5 pts	UN	5 (26%)	UN	4/5 (80%)	1/4 (25%)	UN	6–23 months
Harmon et al. (2016) [28]	45	12/45 (27%)	38 (84%)	LVEF ≤35% or pacing indication and LV GLS ≥−15% and/or NSVT/frequent PVBs with syncope or planned HTX	<50% in 31 pts <35% in 14 pts	2 (4%)	12 (27%)	2 (4%)	11/12 (92%)	UN	27/45 (60%)	17 ± 14 months
Chuzi et al. (2018) [33]	31	14/31 (45%)	25 (80%)	Not specified	43 ± 14%	UN	2 (6%)	2 (6%)	2/2 (100%)	UN	19/31 (61%)	15 ± 11 months
Rezc et al. (2018) [33]	15	15/15 (100%)	14 (93%)	NSVT and syncope/presyncope	53%	4 (27%)	4 (27%)	UN	3/4 (75%)	2/4 (50%)	13/15 (87%)	49 months (median)
Kim et al. (2019) [30]	23	7/23 (30%)	23 (100%)	LVEF ≤35% or NSVT and/or syncope	36 ± 14%	UN	6 (26%)	1 (4%)	6/6 (100%)	0/6 (0%)	14/23 (61%)	3.24 years (median)
Donellan et al. (2019) [37]	38	0/38 (0%)	35 (92%)	Not specified	-	UN	8 (21%)	UN	UN	2/8 (25%)	UN	42 ± 26 months
Higgins et al. (2020) [34]	472	UN	356 (75%)	Not specified	≤30%in 236 pts; >30–40% in 99 pts; <40% in 119 pts	116 (25%)	UN	UN	UN	UN	345/472 (73%)	42 months (median)
All studies	718	118/246 (48%)	569/718 (79%)	-	-	128/554 (23%)	57/246(23%)	13/174 (7%)	30/34 (88%)	6/27 (22%)	449/661 (68%)	-

CA—cardiac amyloidosis; HTX—heart transplantation; ICD—implantable cardioverter-defibrillator; LVEF—left ventricular ejection fraction; LV GLS—left ventricular global longitudinal strain; UN—unknown; NSVT—non-sustained ventricular tachycardia; pts—patients, PVBs—premature ventricular beats.

## Data Availability

No new data were obtained in this study.

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
