# Peer review of "Arrhythmic Sudden Cardiac Death and the Role of Implantable Cardioverter-Defibrillator in Patients with Cardiac Amyloidosis—A Narrative Literature Review"

_jcm, 2021, doi:10.3390/jcm10091858_

Round 1

Reviewer 1 Report

The authors attempt to give a narrative but comprehensive overview of the role of ICD therapy in patients with cardiac amyloidosis. However, I feel that the authors have not included all the relevant and predominantly most recent literature on this topic. Therefore, either the search was not conducted properly or selection was somehow incomplete.

Please include the following MS (pmid) for the following parts of the MS:

-Introduction:

pmid  31359320, 31912620. Please also correct reference 2 as it should be "Jacc CardioOncology'.

-M&M: why do the authors refer to reference 7 when describing the search? given the not up-to-date search, is it possible that the authors used the same approach as ref 7 which means literature up to 2018 was included and not further?

-prognostic factors:

*why do the authors mainly focus on risk stratification of patients with AL-CA? please also describe risk prediction in ATTR-CA.

*although the authors describe EP studies on mechanisms of ventr arr, it is very preliminary. Please eludicate more in detail and also include the role of cardiac imaging (MRI, edema, ECV) etc in terms of substrate analysis and its role in prognosis.

-ICD therapy:

this section needs to be restructured significantly, adding the following manuscripts (pmid)

32514579,    33088185, 32327068 (please look carefully at the table on ICD studies)

I miss an author's interpretation:

what do the authors think is the best strategy? or stick to a guideline? what are the considereations?

a central illustration would greatly enhance the paper.

Reviewer 2 Report

This is only a narrativ and not - as would be absolutely necessary in my view - a systemic review on a complex topic.

One study comprises the vast majority of the data (472 out of 680 patients included). 

The lack of reliable available data renders any valuable conclusion of the review impossible impossible (maybe with the already well-known fact that large and well-done prospective studies are clearly needed in patients with cardiac amyloidosis).

Reviewer 3 Report

Dear Editor

The manuscript entitled “Title Arrhythmic sudden cardiac death and the role of implantable cardioverter-defibrillator in patients with cardiac amyloidosis – a narrative literature review” (jcm-1144382) presents a narrative review about the possible role of an implantable cardioverter-defibrillator (ICD) for primary prevention of sudden cardiac death in patients with cardiac amyloidosis.

The manuscript provides a narrative review of available studies, which are limited and not conclusive. However, I think the topic is of interest to promote studies on pathophysiology of cardiac amyloidosis and about the electrophysiological and structural modification of the heart tissue.

I suggest explaining criteria for the selection of the 28 out of 1734 records, then it is not clear to me if the Authors further focused on the 10 studies reported in table 1 or if alle the 28 records are discussed in the manuscript. In this case, if 10 studies .

I also suggest a language revision of the manuscript, there are several typos.

Best regards

Round 2

Reviewer 1 Report

All the points raised in the previous review round have been addressed adequately. I have no further comments.

Reviewer 2 Report

This narrative review states that current studies are not sufficient to give any recommendation about ICD use in primary prophylaxis in amyloidosis. Therefore, its impact is rather limited.